# Adolescents, Ambivalent Sexism and Social Networks, a Conditioning Factor in the Healthcare of Women

**DOI:** 10.3390/healthcare9060721

**Published:** 2021-06-12

**Authors:** Jose Luis Gil Bermejo, Cinta Martos Sánchez, Octavio Vázquez Aguado, E. Begoña García-Navarro

**Affiliations:** 1Department of Sociology, Social Work and Public Health, University of Huelva, 21004 Huelva, Spain; cmartos@dstso.uhu.es (C.M.S.); octavio@dstso.uhu.es (O.V.A.); 2Research Group “Estudios Sociales e Intervención Social”, University of Huelva, 21004 Huelva, Spain; esperanza.garcia@denf.uhu.es; 3Department of Nursing, University of Huelva, 21004 Huelva, Spain

**Keywords:** sexism, social networks, adolescence, digital gender gap, emotional well-being

## Abstract

Even though gender equality being present in the social and political sphere, we still encounter aspects that are characteristic of sexism. Such aspects impact upon gender inequality and different types of violence towards women. The present article aims to examine the behaviour of adolescents from Huelva with regards to ambivalent sexism towards women on social networks and their influence on health. Furthermore, we seek to uncover adolescent’s perceptions with regards to gender differences in the use of social networks, the relationship between sexism and women’s emotional well-being was observed. The study sample was formed by young people aged between 14 and 16 years who were residing in rural and urban zones in the south of Spain. A mixed methods approach was taken. At a quantitative level, a sample of 400 young people was recruited. These were administered a questionnaire about sexism which was composed of two scales and has been validated at a national and international level. At a qualitative level, the study counted on 33 young people who participated in in-depth discussions via interviews and discussion groups. The results showed that sexism emerges in adolescence in the analysed sample from the south of Spain. This favoured a digital gender gap and was reinforced through social networks such as Instagram and Snapchat. Rising awareness and a critical view of the aforementioned sexism was shown on the behalf of females, particularly those from urban backgrounds.

## 1. Introduction

In a postmodern society, such as today’s, gender equality is seemingly addressed by public authorities and assumed to exist in society in general [1,2,3]. Steps towards gender equality and women’s freedom have supposedly challenged the hegemony of the patriarchal system [4]. Yet, we maintain the belief that gender inequality still persists to a large extent and often manifests itself as recurrent violence towards women [5]. Unfortunately, such violence is being observed at increasingly younger ages, even being seen within the adolescent population [6]. This reality motivated us to conduct the present research.

Sexism is a complex construct that forms a part of the gender differentiation between males and females. It has especially negative repercussions for females and for anyone who moves away from the dominant hegemonic masculinity [7]. Sexism in itself, houses a set of beliefs regarding respect for established gender roles. It has very negative mental health repercussions for women and perpetuates their subordination to men [8,9,10]. A nuance of this can be explored through the concept of ambivalent sexism [8], which is composed of masculine dominance and represented through hostility and interdependence. From this perspective, ambivalent sexism has two components, namely, hostile sexism and benevolent sexism [8,9,11]. Hostile sexism concerns a gender ideology that is directly discriminatory and violent towards women. It has three fundamental factors. The first is male paternalism towards females in the sense of dominance due to the belief that women are the weaker sex relative to men. The second factor pertains to a competitive gender differentiation in which the belief is held that women cannot take responsibility over important issues, such as those that are economic or social in nature, in the same way that men can. The final factor concerns heterosexual hostility and assumes that women present a danger and a manipulative force to men [12]. On the other hand, we find that which is denominated benevolent sexism. This ideology involves subtle gender discrimination and is characterised by the following factors: Paternalism, complementary gender differentiation and cisgender heteronormative intimacy. The former is manifested in this type of sexism from the standpoint of protection, whilst the second factor refers to positive characteristics which complement the man and, equally, mark gender differences in a supposedly logical way. The latter, cisgender heteronormative intimacy comes from the belief that a man is incomplete without a woman [8,10].

During adolescence, we can observe that ambivalent sexism is transmitted through social networks. This occurs in a number of ways [13], for instance in writing via hashtags, likes, videogames, music and images, in addition to through expressive instrumentalised means, above all through mobile phones [14]. In the adolescent population, this type of sexism, through social networks, takes on special characteristics with regards to the rapid and direct dissemination of material, which quickly becomes viral in adolescent groups [15]. On the other hand, the anonymity offered by social networks means they can be used to promote more violent forms of sexism towards women [16,17]. The tendency towards audio visual preferences amongst younger people as a form of expression on social networks, for instance through photographs or videos [18,19,20], shows that females present such images in a more sexualised way. This occurs in virtual daily practices such as that known as sexting [21,22,23] and reflects sexism against women and the risk posed by this type of practice. Worryingly, such practices are increasingly more common on social networks, especially amongst women [24]. For this reason, it is important to deepen the knowledge of this worrying and current issue that especially affects the emotional state and healthcare of younger women [25]. The aim of the present work was to observe the way in which ambivalent sexism manifests itself through social networks in both men and women, a situation that has been shown to particularly affect the emotional health of women. Furthermore, we sought to examine its impact within a 14- to 16-year-old population from the region of Huelva given that this is an important age for determining the future state of the adult population and this geographical region is characterised by both urban and rural areas which have very different social realities.

The present study contributes to the examination of gender theory in relation to women’s mental health through an analysis of the relationship of variables related with sexism and social networks. The gender theory perspective leads us to gender inequality in adolescence, expressed through sexism and social networks, since these issues have an impact on the emotional wellbeing of people, especially women. The key contribution of the present research is, therefore, its implications when it comes to addressing inequality between men and women [9,13,15].

## 2. Method

In order to respond to the proposed objectives, the present work took a quantitative and qualitative methodological approach which was fed by a pluralist methodological approach. It was accompanied by data and information triangulation [26]. This will enable us to better approach subtle issues regarding ambivalent sexism and gender differences [27,28]. Research met the requisite ethical conditions by obtaining informed written consent from minors’ legal guardians.

### 2.1. Participants

A sample of 400 students aged between 14 and 16 years was selected from public educational centres in the province of Huelva. All participants were compulsory secondary education (ESO) students and came from 7 public educational centres. Students were classified as coming from rural (>5000 inhabitants) or urban (<5000 inhabitants) settings. The average age of participants was 15.01 years (SD = 0.82). Non-random sampling was performed in consideration of the study categories that is was necessary to represent. In other words, participants were selected to represent males (n = 200) and females (n = 200), whilst also equally representing all ages between 14 and 16 years, and rural and urban geographic regions (see Table 1). Survey data was only included from fully completed surveys with incompletes surveys being discarding.

With regards to the qualitative analysis, the sample was formed of three discussion groups. Of these, one was formed by 7 males, another was formed by 8 females and the third was a mixed group of 10 individuals. In addition to this, 8 in-depth interviews were conducted with four females and four males. Interview participants were different to those who attended discussion groups. Overall, qualitative analysis included a total of 33 individuals from urban and rural settings.

### 2.2. Instruments

Two types of rating scales on ambivalent sexism were administered. Some sociodemographic data were also collected, alongside to questions about social networks. The scales are described below.

The ambivalent sexism inventory, ASI, developed by Glick and Fiske [8] in the reduced Spanish version of Rodríguez, Lameiras and Carrera [29]. with hostile (traditional) and benevolent sexist attitudes (positive affective tone) towards women are measured. All items have a 6-point response format (from 1 “totally disagree” to 6 “totally agree”). Higher scores indicate higher level of sexism towards women. The present study obtained a Cronbach reliability alpha of 0.81 for hostile sexism (HS) and 0.76 for benevolent sexism (BS). The inventory of ambivalent sexism consists of 11 items and is intended for the general population.

The Detected Sexism in Adolescent scale (DSA), the most updated version [30], validated in its smallest version [31]. This scale measures in their items’ hostile sexism and benevolent sexism (ambivalent sexism). The articles were answered on a 6-point Likert scale, with options running from 1 (totally disagree) to 6 (totally agree). The 10-item scale is intended for the adolescent population and measures ambivalent sexism. Two scales of ambivalent sexism were required in order to measure the hostile and benevolent aspects of sexism. This is important in order to be able to later link ambivalent sexism with social networks and their subsequent impact on women’s emotional health.

In addition to including sociodemographic data for the instruments, we added two questions in relation to social networks in order to be analyzed with the previous scales. The first is, which networks are most commonly used? This question was posed with multiple response options being provided, specifically, the following alternatives were given: Whatsapp, Twitter, Facebook, Youtube, Tuenti, Instagram, SnapChat, blogs and other social networks. Whatsapp was included although it is not really a social network because we consider that this instant messaging application increasingly possesses characteristics that are fitting of a social network at its broadest communication level (through groups, distribution lists and 24 h states). The second question strives to uncover the reasons for which these networks are used. As with the former question, response options allowed various responses to be given, including: to gossip about others, talk with my partner (where relevant), communicate with my family, for school use and learning, to meet others, to hook up and to talk with friends/people I already know.

Qualitative analysis produced the following preliminary categories: differentiated use of social networks, references to the type of social networks and reasons for their use and attributes associated with the identification of either of the genders. Different categories underlying ambivalent sexism emerged during the interviews and discussion groups. Such themes arose following reflection.

### 2.3. Procedure

An observational design was developed, which was descriptive and cross-sectional in nature. The sample was selected through non-random convenience sampling. This was made possible sue to the ease of access granted by the participating educational centres in the province of Huelva. When selecting educational centres, the number of inhabitants in the locality was taken as a reference. Urban and rural settings were defined as having more or less than 5000 inhabitants, respectively. We selected a total of 7 public educational centres, with 4 coming from urban settings and 3 from rural settings. Once access was granted by the centre, a date and time was agreed upon to complete questionnaires. We then sent informed consent forms to the educational centres for legal guardians to sign on behalf of the students. The educational levels examined corresponded to the 3rd and 4th years of ESO. On the day on which questionnaires were completed in the classroom, discussion groups and interviews were organised with the students who voluntarily agreed to participate in one of the two slots. Volunteers were recruited until sex, age and geographical location groups were all well represented.

Qualitative analysis techniques were carried out the following week. Two team members participated in the interviews with the aim of minimizing bias resulting from not having physical contact between agents. One researcher proceeded to conduct the interview whilst the other took over the technical aspects and took fieldnotes.

### 2.4. Data Analysis

Quantitative data analysis was conducted using the statistical analysis program SPSS (IBM Statistics v.25). Comparative analysis was conducted of basic variables (n = 400). The variables explored were ambivalent sexism, use of social networks and sex. The main dependent variable was sexism and Chi-square analysis was performed.

Qualitative analysis was performed based on discourse analysis, considering emergent discourse from both discussion groups and interviews. This was conducted using the qualitative data handling program ATLAS. Ti. V.8. The categories that resulted from this analysis were as follows: differentiated use of social networks, references to the type of social networks and reasons for their use, and attributes associated with the identification of either of the genders. Next, the categories and sub-categories to emerge from quantitative analysis following the aforementioned techniques are detailed (see Table 2).

During data collection, informed consent was received from informants and confidentiality and anonymity were maintained throughout. Prior to study start, participants were informed about the study objectives and the bioethical principles of the Helsinki declaration were respected. In addition, data obtained from the various discourses comply with current regulations regarding the protection of personal data.

## 3. Results

Next, we describe outcomes, giving detailed information on the extent of associations between the two used scales (ASI and DSA) in the adolescent population. It is worth mentioning that both scales measure ambivalent sexism but according to the different categories of hostile and benevolent sexism. The present study considers sexism in general when referring to ambivalent sexism. Outcomes presented in the following tables pertain to sexism and use of specific social networks (Instagram, Snapchat, YouTube/Blogs).

With regards to the study objectives, the existing relationship was analysed between the two ambivalent sexism scales employed and the social networks Instagram and Snapchat. Two scales were used that pertained to ambivalent sexism in adolescents (DSA) and ambivalent sexism in the general population (ASI), with these scales being significantly correlated (*p* < 0.001). Mean scores of the two scales are classified as sexist or otherwise with 57.6% of adolescents reporting scores corresponding to sexism. Chi-squared outcomes showed no significant differences between males and females.

A relationship was observed between sexism and Instagram use, with a significant relationship (*p* < 0.05) emerging on both scales, although a particularly strong relationship emerged for 8 items of the DSA scale and 3 items of the ASI scale (see Table 3).

If we move on to the social network Snapchat, we find more relationships between the variables, with these associations also showing a higher significance level (see Table 4). With regards to the DSA scale, for 7 of the sexist beliefs a positive association was found between confirmation of sexism and using Snapchat, with the same outcome for young people who used the social network and stated sexist beliefs. With regards to the ASI scale, more relationships emerged in relation with Snapchat than with Instagram, with significant relationships emerging with 10 sexist beliefs in the case of the latter.

As we can see in Table 4, we observe that the use of other social networks, such as YouTube and blogs, was also related with sexist beliefs. In this case, aspects such as perceiving there to be greater compassion and suffering amongst women were related with the use of YouTube, whilst the idea that women should be put on a pedestal by men or that women have greater moral sensitivity was related with blog use. On the other hand, some results also pointed to the lack of associations. For instance, users of Tuenti did not demonstrate a large extent of sexism, whilst at the same time, those who did not present with sexist beliefs tended not to use Tuenti. Specifically, this relationship emerged with regards to beliefs around raising children, the fragility of women in respect to men and the idea that they should be rescued by men (see Table 5).

Here we finish the presentation of the associations uncovered between data pertaining to social networks and sexist beliefs. Without a doubt, the two social networks with more visual impact such as Instagram and, especially, Snapchat, were the networks most strongly related with sexist beliefs when measured on either of the two utilized scales. We will now move onto the qualitative aspect in order to better understand the nuances present with regards to the relationship between sexism and social networks (see Table 6).

With regards to the discourse analysis (Transcription of categories: [UW16]: 16-year-old urban woman; [RM15]: 15-year-old rural man.), we will begin by addressing the relationship between sexism and sex.

Within women, we observe the way in which images of feminine ideals are promoted, not only with regards to attributes but also the sexist beliefs that seem to be acknowledged by women and act as coercive means to restrict behaviour within what is accepted by established norms.

−“Girls who look like boys are not viewed well, in fact they tell you that you are a lesbian, you’ve got to uphold a certain feminine image, you can be hooking up with another girl, but whilst you don’t hold yourself like a guy there isn’t any problem, nobody messes with you, if you always look good [UW15]”.−“It’s not that we are better, more charming, more patient, what happens is that they label us in that way and if you don’t abide by that even just a little they let you know [RW16]”.−“Here in the village the control is incredible, any situation, the way you dress or who you hang out with marks you and conditions you, for this reason you have to know really well what you are doing if you don’t want to mess up [RW14]”.

In the accounts given by females from rural settings social control emerged to a greater extent than it did in the accounts of urban females, where this theme emerged occasionally. Without a doubt, in settings with fewer inhabitants, such as in rural environments, there seems to be a greater indication that behaviour occurs outside of that which is normally accepted, in this waypromoting hostil sexism. In contrast, in urban areas there is a greater sense of criticism by females towards the recognition of hostile sexism through behaviour.

−“Yeah, but what do you do? On the one hand you have to be someone who waits on the man and lets him take decisions, and you’re there as if you weren’t, as if you didn’t know what to do, stay or go, get out of the way, it’s a pain, sometimes I can’t be bothered with it and I do what I like, and let them say that I’m losing my mind [UW14]”.−“For me, when its best for me I act one way and when it isn’t I act another, I watch and act accordingly [UW14]”.−“I think that this doesn’t happen in relationships between us, I don’t feel like I have to play a role for them, you are more you, you’ve got to go outside of the norm a bit, if you don’t you will stay in the last century [UW16]”.

To another extent, in the discourse of males we see that “being a gentleman” is more associated with chivalry in males from rural areas. In this sense, it emerges as a form of masculine identity that is inherent to benevolent sexism.

−“Being attentive and a gentleman with girls is appreciated, as they give us attention and care for us better in a way that another person probably won’t [RM14]”.−“The quality of a person is shown in the small details, and girls need affection and to be waited on, they like these things, we don’t care so much about that, but for them it is important [RM16]”.−“I see my father do it and I don’t see why it is wrong I don’t know why it is criticized, it is being polite [RM14]”.

Another dominant discourse to emerge amongst females, in this case amongst both rural and urban females, is the belief that women complement men. In this way, a heterosexual system of relationships and couples is promoted:−“Without us they are lost, they don’t know how to do anything, we have all the power [UW14]”.−“As hard as they try, with their buddies, going out, drinking with friends, wherever, they always come to find us afterwards, even if it is just to look good in front of the rest of their group, they need us [RW16]”.−“When they settle down later they come looking for you, deep down with us they share their most personal issues, we never leave them hanging, if they know how to behave [UW16]”.

Another key aspect that we see in the discourse and represents a characteristic of benevolent sexism is the image of females. Stereotypes are associated to the female gender with regards to their behaviour, beauty and expression, and the way this is transmitted through social networks. In this sense, certain images are demanded by males and, resultantly, taken on by females:−“I fix myself up as required, and when I do it I reap the rewards, I get myself out on social networks as best I can, with my selfies they don’t confuse me with any old village lowlife [RW14]”.−“They should see you smile, we have to be ready and always prepared on the networks, to a high level [RW16]”.−“Women need a bit more loving on social networks, they are more tender, I definitely bear it in mind, obviously I like to see a beautiful girl and I will follow her on the networks [RM14]”.−“It seems crazy to me but I recognise that I log in and I look, I like to see beautiful girls, with good style doing selfies, if the photo is tacky I’m not interested [RM16]”.

## 4. Discussion

Various recent studies have verified ambivalent sexism to be a reality for Spanish youth [32,33,34]. In our study, males score higher in hostile sexism [35] despite its low social desirability [36].

Through the data obtained in the present research we identified equally high levels of sexism in both sexes. The qualitative nuance showed that females recognise sexism as a form of coercive social conditioning, which is manifested as behaving within accepted norms, a factor that generates hostile sexism [37].

In the same way as in the present research study, recent studies present a relationship between sexism in adolescents and social networks [38,39].

Another increasingly questioned aspect is the image of a chivalrous man, a characteristic that is framed within benevolent sexism. For women, this concept concerns education and is not associated with gender as they express not desiring to receive a different treatment simply for being women. For some men, this concept related to an identity pattern that pertains to how a man expresses interest in a woman in heteronormative relationships [40].

With regards to the examined differences in sexist beliefs between adolescents in rural and urban environments, outcomes did not reveal significant differences (*p* > 0.495). In contrast of this, the discourse analysis revealed that feelings of control in women are more important in the rural context as rural women explicitly referred to the risk they run by refusing to comply with accepted norms. Furthermore, chivalry was also observed to be an increasingly valued trait in the rural setting, especially for men. In some cases, it was even perceived as a model of masculinity to be followed within heterosexual couples by providing a way to act on romantic feelings [41].

The social networks found to be most strongly related with sexist beliefs were Instagram and Snapchat. Curiously, both encourage the use of photographs and videos. The influence of images, reflected by Instagram and Snapchat, shows us that the two social networks most related with sexism are characterised by various image-related features. These include a limited display time, almost instantaneous speed of content transmission and risk-taking in the exposure of images and text. The latter opens users up to an environment where one is observed and can act according to their free will without any restrictions [42].

With respect to photographs presented through *selfies* in females, these represent an image of a person who is alone in front of the camera and observes how the photo will look whilst they take it. In this way, individuals can present the image they wish to give within virtual settings. Furthermore, they later have the opportunity to fulfil idealized and sexualized desires, which are generally masculine [43,44]. This puts women in a situation in which they are more reliant on group self-esteem and this makes their mental health more vulnerable. Both numeric data and discourse showed us that sexism and social networks leads us to present and promote images of the ideal woman, especially, in relation to beauty ideals or formats of established femininity. Within this, we find ideals of sweetness, indulgence, sensuality, sexuality and many other attributes pertaining to how a woman should be. The image of women in social networks, given through photos and videos, is highly related with sexist beliefs. In such settings, women are once again particularly exposed as they are more vulnerable than men. Such settings can result in situations of harassment, blackmail and violence, in this way, opening up a new digital gender gap [45,46].

For the adolescents of Huelva, we can observe that social networks are not only associated with the digital gender gap. In fact, new and contrasting alternative performativity’s are also emerging, which are more critical in nature and demonstrate disagreement with gender inequality in youth populations [47,48], all of this has repercussions on women’s emotional distress, and their mental health has deteriorated to a great extent [49]

## 5. Conclusions

Results demonstrate that sexism remains an existing reality in adolescence, regardless of gender, with women being more aware of sexism and of the stigmatising factor women face when breaking free of norms and expected behaviours. In rural settings, this is manifested as a strong sense of control due to having to behave in accordance with sexist patterns in order to be socially accepted, a determinant of hostile sexism.

With regards to the heteronormative beliefs of establishing partnerships that are ideally between men and women, diverse sexist beliefs were upheld by both sexes, especially in rural settings. Another factor to be considered is that of chivalry, a characteristic of benevolent sexism. Whilst in men, this is occasionally established as a way of cementing identity, for example in the way a man expresses his love for a woman, this construct is increasingly less important within women and sometimes even criticised as an undesirable trait in men.

The social networks to most represent sexism were Instagram and Snapchat. This suggests that it is especially important for women to present themselves by sending images and videos with the aim of these becoming viral. Such acts respond to stereotyped feminine ideals, which revolve around ideas of dominant masculinity. This can be seen in the specific case of selfies that expose women to vulnerable situations in which their image is put at risk in a way that could seriously affect their self-esteem and emotional wellbeing and lead them to suffer gender-based violence. It is of great importance that future research continues to examine the way in which sexism occurs through specific social networks, photographs, videos, text, etc. Furthermore, we feel that it is important to investigate other non-heteronormative choices pertaining to relationships or desire, going beyond binary conceptions of sex or cisgender. The aim of this is to be able to be closer to the dynamic and changing reality currently lived by adolescents. On the other hand, the relationship between sexism in social networks greatly influences emotional well-being, especially among women, affecting their mental health.

In contrast of the digital gender gap related with existing sexism, we see that new standpoints, emerging masculinities, feminisms and forms of expression on social networks, show subjective, alternative, reflective and critical ways of thinking, which fall outside of the dominant behaviours or beliefs of youth.

## Figures and Tables

**Table 1 healthcare-09-00721-t001:** Sample: Age, sex and geographical location.

Categories	Geographical Location		Total
	Urban	Rural	
14 years	100	32	132
15 years	96	38	134
16 years	85	49	134
Men	144	56	200
Women	137	63	200

**Source**: Developed by the author.

**Table 2 healthcare-09-00721-t002:** Categories and subcategories to emerge from the study according to different techniques.

Dimension	Categories	Subcategories	Interviews	Discussion Groups
Ambivalent sexism on social networks	Differentiated use of social networks between males and females	Stereotyped image given of women	x	x
		Social networks used	x	x
	Benevolent sexism on social networks	Feminine stereotypes	x	x
		Chivalry		x
	Hostile sexism on social networks	Stigmatisation towards women if they don’t conform with feminine stereotypes		x
	Geographic setting	Rural (control)		x
		Urban (critical view)		x

**Source**: elaborated by the authors in relation to data obtained through qualitative analysis.

**Table 3 healthcare-09-00721-t003:** Relationship between the social network Instagram and sexism.

Variables	% of Young People Who Present with Sexism and Use Instagram	Chi-Squared Value
Instagram and ‘patient woman’ **	82.5%	10.1
Instagram and ‘tender woman’ ***	82.7%	14.48
Instagram and ‘accommodating woman’ ***	84.6%	16.52
Instagram and ‘sympathetic woman’ ***	83.5%	13.1
Instagram and ‘fragile woman’ **	84.5	11.38
Instagram and ‘forgiving woman’ **	81.7%	5.73
Instagram and ‘woman at home’ **	81.1%	4.57
Instagram and ‘women suffering’ **	82.3%	7.02
Instagram and ‘women as a complement’ **	88.4%	6.44
Instagram and ‘loved and protected woman’ **	81%	6.12
Instagram and ‘man without woman’ **	81.9%	8.51

Note: ** *p* < 0.05, *** *p* < 0.001; degrees of freedom = 1, DSA scale (gray), ASI scale (dark gray). Source: Developed by the authors.

**Table 4 healthcare-09-00721-t004:** Relationship between the social network Snapchat and sexism.

Variables	% of Young People Who Present with Sexism and Use Snapchat	Chi-Squared Value
Snapchat and ‘tender woman’ ***	72.8%	24.1
Snapchat and ‘sympathetic woman’ ***	73.4%	19.13
Snapchat and ‘women raise children’ **	70.7%	10.16
Snapchat and ‘forgiving woman’ ***	72.3%	12.79
Snapchat and ‘fragile woman’ ***	47.1%	26.1
Snapchat and ‘sensitive woman’ **	69.2%	5.68
Snapchat and ‘woman in the home’ **	70.6%	7.99
Snapchat and ‘women complement men’ ***	70.6%	17.12
Snapchat and ‘other heterosexual sex’ **	69.3%	4.72
Snapchat and ‘woman’s purity’ **	68.7%	4.27
Snapchat and ‘loved woman’ **	69.2%	6.67
Snapchat and ‘love between man and woman’ **	69.4%	8.93
Snapchat ‘man without a woman’ **	70.1%	8.66
Snapchat and ‘women on a pedestal’ **	69.9%	4.9
Snapchat and ‘moral woman’ **	68.8%	7.2
Snapchat and ‘accommodating woman’ ***	74.6%	22.35
Snapchat and ‘patient woman’ **	71.1%	11.41

Note: ** *p* < 0.05, *** *p* < 0.001; degrees of freedom = 1, DSA scale, (gray), ASI scale (dark gray). **Source**: Developed by the authors.

**Table 5 healthcare-09-00721-t005:** Relationship between other social networks and sexism.

Variables	% of Young People Who Present with Sexism and Use YouTube/Blogs	Chi-Squared Value
YouTube and ‘sympathetic woman’ **	69.2%	4.27
YouTube and ‘women suffering’ **	68.2%	5.76
YouTube and ‘women on a pedestal’ **	67.6%	6.88
Blogs and ‘moral woman’ **	69.8%	4.53
Variables (Inverse Relationship)	% of Young People Who Do Not Present with Sexism and Do Not Use Tuenti	Chi-Squared value
Tuenti and ‘women raise children’ **	80.1%	8.41
Tuenti and ‘fragile woman’ **	77.2%	6.19
Tuenti and ‘rescued woman’ **	76%	5.74

Note: ** *p* < 0.05, *** *p* < 0.001; degrees of freedom = 1, DSA scale, (gray), ASI scale (dark gray). Source: Developed by the author.

**Table 6 healthcare-09-00721-t006:** Categories, subcategories and narratives associated with benevolent sexism in the adolescent population of Huelva (qualitative analysis).

Dimension	Categories	Subcategories	Relates
Sexism ambivalent on social networks	Differentiated social network use between males and females	Gender stereotyped images are stronger for females Use of social networks	− “Girls who look like boys are not looked well upon, they tell you that you are lesbian, you have to keep a certain feminine image, you can be hooking up with another girl but, whilst you don’t present yourself like a guy there is no problem, nobody messes with you, if you look good in life [UW]”. − “I fix myself up just enough, and when I do I reap the rewards, I put myself out online as well as I can, with my selfies they don’t confuse me with any village lowlife [RW14]”.
	Benevolent sexism on social networks	Feminine stereotypes Chivalry Heteronormativity	− “Being attentive and a gentleman with girls is appreciated, they give you attention and take care of you when another person probably wouldn’t [RM14]”. − “They [the guys] are lost without us, they don’t know how to do anything, we have all the power [UW14]”.
	Hostile sexism on social networks	Stigmatization towards women if they don’t conform with feminine stereotypes	− “It’s not that we are better, more charming, more patient, it’s just that they give us that tag and as little as you don’t conform with it they pull you up [RW16]”.
	Geographic setting	Rural (control)	− “Control here in the town is incredible, whatever situation, way of dressing or who you go around with marks you and they condition you, for this you have to know well what you do if you don’t want to mess up [RW14]”.
		Urban (critical view)	− “I think that in the relationships between us this doesn’t happen, I don’t feel that I have to play a role for them [boys], you are more you, you have to break with the norm, if you don’t you will stay in the past century [UW16]”.

**Source**: Developed by the author.

## Data Availability

The data is held by the research team and will not be published due to data protection law, but can be consulted on request.

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
