# Peer review of "Adolescents, Ambivalent Sexism and Social Networks, a Conditioning Factor in the Healthcare of Women"

_healthcare, 2021, doi:10.3390/healthcare9060721_

Round 1
Reviewer 1 Report
The topic of the manuscript “Adolescents, ambivalent sexism and social networks, a conditioning factor in the healthcare of women” is interesting. However, the manuscript is, overall, very confusing and has serious flaws.
Introduction
In the introduction the problem should be framed more broadly, reviewing the literature to convey to readers the scope of the problem, its contexts, and its implications. The relevance of each study variable should be clearly described and explained.
Since ambivalent sexism and social networks, and their conditioning factor in the healthcare of women are named in the title, such variables and the relationship between them should be addressed in the introduction, as well as in the methodology, results, and discussion sections.
The problem under investigation is unclear. Also, in the Abstract on page 1, lines 11-13 says "The present article aims to examine the behaviour of adolescents from Huelva with regards to ambivalent sexism towards women on social networks and their influence on health". The reason why the article aims to examine the behavior of adolescents from Huelva should be explained, as well as the reason why adolescents between 14 and 16 years old are chosen.
Method
The method section should include the conceptual and operational definitions of the variables used in the study. The questionnaires used in the study should be explained clearly and completely, including data on the number of items that make up each of the two questionnaire scales (hostil sexism and benevolent sexism) and their psychometric properties.
It should also explain why two questionnaires are used that seem to assess the same variables: hostile sexism and benevolent sexism.
Important contradictions are observed in the sampling procedures: On page 2, lines 86-87 says “A representative sample of 400 students, aged between 14 and 16 years, from public educational centres of the province of Huelva was selected”. And on page 3, lines 131-132 says “The quantitative sample was selected through non-random convenience sampling”. Also, on page 3, lines 141-142 says “discussion groups and interviews were organised with the students who voluntarily agreed to participate in one of the two slots”. It is difficult to understand how a sample that has been selected as described on page 3 can be “A representative sample”.
In Data Analysis (page 4, line 148) it says that “Quantitative data analysis was conducted using the statistical analysis program PSPP”. You should put the complete reference of the “PSPP” program.
Results
The results section is very confusing. Among the problems, it stands out that there is talk of “% of young people who present with sexism” but it is not known what type of sexism is referred to since the instruments used assess Benevolent sexism and Hostile sexism.
The results of Tables 2 (page 5), Table3 (page 6) and Table 4 (Page 7) are not understood. Thus, for example, in table 2, the first column of results shows “% of young people who present with sexism and use Instagram” and in the second column of results it says ”% of young people who use Instagram and present with sexism”. I do not understand the difference between both variables. And the same happens with Table 3 where it puts in the first column "% of young people who present with sexism and use Snapchat" and in the second "% of young people who use Snapchat and present with sexism". And the same type of problem appears in Table 4.
Author Response
Author's Reply to the Review Report (Reviewer 1)
Dear reviewer, first of all I would like to thank you for the time and contributions you have made to my work, undoubtedly and thanks to them it is substantially enriched, below I present the modifications made by each of the proposals, as well as the page where the contributions are found in the text.
Introduction
In the introduction the problem should be framed more broadly, reviewing the literature to convey to readers the scope of the problem, its contexts, and its implications. The relevance of each study variable should be clearly described and explained.
The context and implication of the study variables are specified in the introduction: (On page 1, lines 38-39; on page 2, lines 74-80).
Since ambivalent sexism and social networks, and their conditioning factor in the healthcare of women are named in the title, such variables and the relationship between them should be addressed in the introduction, as well as in the methodology, results, and discussion sections.
We relate the influence of ambivalent sexism in social networks as a conditioning factor in women's health in the introduction (on page 1, line 39), results (on page 13, lines 353-354) and conclusions (on page 14, lines 384-387).
The problem under investigation is unclear. Also, in the Abstract on page 1, lines 11-13 says "The present article aims to examine the behaviour of adolescents from Huelva with regards to ambivalent sexism towards women on social networks and their influence on health". The reason why the article aims to examine the behaviour of adolescents from Huelva should be explained, as well as the reason why adolescents between 14 and 16 years old are chosen.
We specify the reason for the choice of the population with respect to age and geographical context (on page 2, lines 76-80).
Method
The method section should include the conceptual and operational definitions of the variables used in the study. The questionnaires used in the study should be explained clearly and completely, including data on the number of items that make up each of the two questionnaire scales (hostile sexism and benevolent sexism) and their psychometric properties.
It should also explain why two questionnaires are used that seem to assess the same variables: hostile sexism and benevolent sexism.
We complete the operational and conceptual definition of the questionnaires that measure the variables, as well as explain why these questionnaires are used (on page 3, lines 124-125; 129-133).
Important contradictions are observed in the sampling procedures: On page 2, lines 86-87 says “A representative sample of 400 students, aged between 14 and 16 years, from public educational centres of the province of Huelva was selected”. And on page 3, lines 131-132 says “The quantitative sample was selected through non-random convenience sampling”. Also, on page 3, lines 141-142 says “discussion groups and interviews were organised with the students who voluntarily agreed to participate in one of the two slots”. It is difficult to understand how a sample that has been selected as described on page 3 can be “A representative sample”.
In the quantitative sample: We clarified the sample selection method, being a non-random convenience sample, determining its relevance through a proportional age selection between men and women (on page 3, lines 100-103).
In the qualitative sample: we selected by age, sex and origin (rural and urban) a representative sample of each area (on page 4, lines 154-155,165-167).
In Data Analysis (page 4, line 148) it says that “Quantitative data analysis was conducted using the statistical analysis program PSPP”. You should put the complete reference of the “PSPP” program.
We add the complete reference of the statistical data analysis program SPSS (IBM Statistics v.25) (on page 5, lines 174-175).
Results
The results section is very confusing. Among the problems, it stands out that there is talk of “% of young people who present with sexism” but it is not known what type of sexism is referred to since the instruments used assess Benevolent sexism and Hostile sexism.
The type of sexism (ambivalent) is specified (on page 5, lines 195-198).
The results of Tables 2 (page 5), Table3 (page 6) and Table 4 (Page 7) are not understood. Thus, for example, in table 2, the first column of results shows “% of young people who present with sexism and use Instagram” and in the second column of results it says ”% of young people who use Instagram and present with sexism”. I do not understand the difference between both variables. And the same happens with Table 3 where it puts in the first column "% of young people who present with sexism and use Snapchat" and in the second "% of young people who use Snapchat and present with sexism". And the same type of problem appears in Table 4.
We clarify the difference of the first and second column in tables:2, 3 and 4, where for example: in Table 2, the % of young people who present sexism and use Instagram: refers to Instagram users within the group of sexist young people, and the % of young people who use Instagram and present sexism, refers to the group of young people who use Instagram the percentage of them who have sexist beliefs, (on page 5, lines 195-204).

Reviewer 2 Report
This paper examines ambivalent sexism towards adolescent women from south of Spain. The methods employed was mixed methods (qualitative and quantitative analyses). The findings can be useful in understanding sexism in adolescence in digital spaces. I think the paper has a lot of strengths such as the data analysis and findings, and the discussion sections. I believe the paper has potential.
However, I have some suggestions for improvement:
In the introduction, it is not clear the original contribution of this study. How does this study extend prior work? How is it different?
On the quantitative data, what is the sex of the sample out of 400? Are these all women, are some men included?
How much missed data occurred? How many completed survey? What is the rate of completion?
Discuss whether participants signed an informed consent form in the procedures/or method for ethical purposes.
On the instruments, provide the alpha coefficient values of the scales.
Work on punctuation issues:
For instance, there should be a comma prior to the word “which”
The period under the last sentence (first paragraph) after 2.Method is missing
In the discussion, add a theoretical contribution to at least one gender/or sexism theory. For instance, you could interpret one of the findings using at theory, or suggest how the findings contribute to theory.
Other than that, I believe the paper can contribute well.
Author Response
Author's Reply to the Review Report (Reviewer 2)
Dear reviewer, first of all I would like to thank you for the time and contributions you have made to my work, undoubtedly and thanks to them it is substantially enriched, below I present the modifications made by each of the proposals, as well as the page where the contributions are found in the text.
In the introduction, it is not clear the original contribution of this study. How does this study extend prior work? How is it different?
The original contribution of this study is specified: (On page 2, lines 81-87).
On the quantitative data, what is the sex of the sample out of 400? Are these all women, are some men included?
We clarify the composition of the selected sample (On page 3, lines 100-103: 112-113).
How much missed data occurred? How many completed survey? What is the rate of completion?
The data on all persons who completed the survey are shown below (On page 3, lines 103-104).
Discuss whether participants signed an informed consent form in the procedures/or method for ethical purposes.
Informed consent and the ethical value of the research are specified (On page 2, lines 93-94; 161-162).
On the instruments, provide the alpha coefficient values of the scales.
The value of the Cronbach's alpha or coefficient alpha, in the two scales used, is reflected (On page 3, lines 122-124).
Work on punctuation issues:
For instance, there should be a comma prior to the word “which”
The period under the last sentence (first paragraph) after 2.Method is missing
Punctuation errors have been thoroughly checked and corrected.
In the discussion, add a theoretical contribution to at least one gender/or sexism theory. For instance, you could interpret one of the findings using at theory, or suggest how the findings contribute to theory.
It specifies how the findings contribute to the theory of gender inequality in the adolescent population (On page 2, lines 81-87).

Reviewer 3 Report
This paper approaches a very interesting and atual issue, considering the problem of sexism, adolescents and their use of social networks. It is well structured and accomplish academic requirements.
The abstract is well done reflecting the content of the whole text. The subject, aims, sample, methods, results and conclusions can be identified.
The introduction starts with the problem and refers to enough (25) and appropriate studies, being most of them recent, that is, published in the last five years. The aim of the study can be found but objectives are not very explicit in the text, despite of being clear in the abstract.
The mixed methodology with triangulation seems well appropiate for the study and it is in accordance with the results presented. The quantitative sample, is called representative, but it should be more detailed, in terms of which population it represents. And if it is a representative sample, inferential statistical tests can be applied, but it was not done. Only Chi-square tests are presented in tables. Moreover, in procedure section, it is referred that “The quantitative sample was selected through non-random convenience sampling.” So, in this condition it is not a representative sample and it should be corrected where is it wrong.
About the sample characteristics, and given the issue of the paper, more factors should characterize the sample, namely, number of boys and girls, how many from urban and rural area. Essentially these two factors or independent variables should be cleared. And a statistical analysis considering them would be appropriate. In terms of data analysis, special atention should be given. The first phrase has several mistakes, namely “PSPP”, when refering to comparative tests (what kind of tests were applied?) and about basic variables (what are these, independent or dependent?).
If a table is presented with the categories for the qualitative part also a table with the complete characterization of the quantitative sample would make sense.
In results section, tables 2, 3 and 4 are a little dificult to understand. The reason to include the two colums with the same variables (ex: % of young people who present with sexism and use Instagram / % of young people who use Instagram and present with sexism) should be clarified. The p value or the notation (*) can stay better near the Chi-squared value.
Qualitative results are presented, but there aren’t a real triangulation between the quantitative and qualitative data of the study. Nevertheless, considering the individual factors presenting in the qualitative analysis, it seems possible to establish a relationship with the quantitative results, and presented clearly.
In the discussion section, for exemple the 5th paragraph (lines 293-297) statements should be supported with results from the two parts, what can be really a triangulation and discussion is the better section to do it.
Some spelling errors are highligthted in the text (in attach) in order to facilitate authors’ revision.

Author Response
Author's Reply to the Review Report (Reviewer 3)
Dear reviewer, first of all I would like to thank you for the time and contributions you have made to my work, undoubtedly and thanks to them it is substantially enriched, below I present the modifications made by each of the proposals, as well as the page where the contributions are found in the text.
The introduction starts with the problem and refers to enough (25) and appropriate studies, being most of them recent, that is, published in the last five years. The aim of the study can be found but objectives are not very explicit in the text, despite of being clear in the abstract.
The objectives of the article are specified in the introduction. (On page 2, lines 74-80).
The mixed methodology with triangulation seems well appropiate for the study and it is in accordance with the results presented. The quantitative sample, is called representative, but it should be more detailed, in terms of which population it represents. And if it is a representative sample, inferential statistical tests can be applied, but it was not done. Only Chi-square tests are presented in tables. Moreover, in procedure section, it is referred that “The quantitative sample was selected through non-random convenience sampling.” So, in this condition it is not a representative sample and it should be corrected where is it wrong.
In the quantitative sample: We clarified the sample selection method, being a non-random convenience sample, determining its relevance through a proportional age selection between men and women (on page 3, lines 100-103).
In the qualitative sample: we selected by age, sex and origin (rural and urban) a representative sample of each area (on page 4, lines 154-155,165-167).
About the sample characteristics, and given the issue of the paper, more factors should characterize the sample, namely, number of boys and girls, how many from urban and rural area. Essentially these two factors or independent variables should be cleared. And a statistical analysis considering them would be appropriate. In terms of data analysis, special atention should be given. The first phrase has several mistakes, namely “PSPP”, when refering to comparative tests (what kind of tests were applied?) and about basic variables (what are these, independent or dependent?).
We specify the characteristics of the selected sample as well as the main variables for data analysis (on page 3, lines 100-103; 112-113; on page 5, lines 175-178).
If a table is presented with the categories for the qualitative part also a table with the complete characterization of the quantitative sample would make sense.
We made a table with the selected sample according to each category (on page 3, lines112-113).
In results section, tables 2, 3 and 4 are a little dificult to understand. The reason to include the two colums with the same variables (ex: % of young people who present with sexism and use Instagram / % of young people who use Instagram and present with sexism) should be clarified. The p value or the notation (*) can stay better near the Chi-squared value.
We clarify the difference of the first and second column in tables:2, 3 and 4, where for example: in Table 2, the % of young people who present sexism and use Instagram: refers to Instagram users within the group of sexist young people, and the % of young people who use Instagram and present sexism, refers to the group of young people who use Instagram the percentage of them who have sexist beliefs, (on page 5, lines 195-204).
Qualitative results are presented, but there aren’t a real triangulation between the quantitative and qualitative data of the study. Nevertheless, considering the individual factors presenting in the qualitative analysis, it seems possible to establish a relationship with the quantitative results, and presented clearly.
During the development of the field work we have carried out data triangulation, since this procedure allows us to obtain greater quality control in the research process and guarantee the validity, credibility and rigor of the results obtained.
In this case that you propose we have added in the manuscript the quantitative data that in this case do not coincide with the qualitative data, due to the differences in the feeling of control of the women in the rural and urban environment, a result that could not have been obtained without the triangulation.
In the discussion section, for exemple the 5th paragraph (lines 293-297) statements should be supported with results from the two parts, what can be really a triangulation and discussion is the better section to do it.
This paragraph specifies the relationship between quantitative and qualitative data. (on page 13, lines 333-340).
Some spelling errors are highligthted in the text (in attach) in order to facilitate authors’ revision.
Spelling errors are carefully checked and corrected.

Reviewer 4 Report
The study covers an important topic---an analysis of adolescent’s perceptions of ambivalent sexism within the context of gender differences in the use of social media networks. The study results can provide a useful contribution to research on the relationship between implicit sexism, digital gender gap and negotiation of heteronormative gender roles on social media. The methodology utilized is sound and appropriate, and the findings are significant. Overall the writing quality is good, the introduction is however weak. For instance, it is not clear what the authors mean by this sentence "Discussion is based on a series of legal, mediatic,
political and social improvements [1,2,3] which supposedly call check-mate on the sexist patriarchal system which has loomed large throughout history". A rewriting of the introduction with editing of the convoluted parts will be good for this paper.
Author Response
Author's Reply to the Review Report (Reviewer 4)
Dear reviewer, first of all I would like to thank you for the time and contributions you have made to my work, undoubtedly and thanks to them it is substantially enriched, below I present the modifications made by each of the proposals, as well as the page where the contributions are found in the text.
The study covers an important topic---an analysis of adolescent’s perceptions of ambivalent sexism within the context of gender differences in the use of social media networks. The study results can provide a useful contribution to research on the relationship between implicit sexism, digital gender gap and negotiation of heteronormative gender roles on social media. The methodology utilized is sound and appropriate, and the findings are significant. Overall the writing quality is good, the introduction is however weak. For instance, it is not clear what the authors mean by this sentence "Discussion is based on a series of legal, mediatic,
political and social improvements [1,2,3] which supposedly call check-mate on the sexist patriarchal system which has loomed large throughout history". A rewriting of the introduction with editing of the convoluted parts will be good for this paper.
The introduction to the research has been rewritten, since it is very important that a topic as sensitive as gender inequality is well understood, as well as the relationship established in the work with the variables that are related (sexism, social networks and adolescent population). On the other hand, the whole text is revised, making some of its parts more understandable.
Thanks for your contribution again, it has been very useful to give a more understandable and dynamic character to the text.

Round 2
Reviewer 1 Report
The revised manuscript "Adolescents, ambivalent sexism and social networks, a conditioning factor in the healthcare of women" shows some improvement compared to the first version. However, the manuscript is still very confusing and had serious flaws. The most important are the following:
- Although authors have included the number of items in the description of the questionnaires, the reliability and validity and how to calculate ambivalent sexism and the percentage of people who present sexism remains unclear.
- I still don't understand the results of Tables 3 (page 7), Table 4 (page 8) and Table 5 (Page 9). The authors have included the following text on page 5, lines 195-204:
“Remembering that both scales measure ambivalent sexism and that within the category of ambivalent sexism there can be hostile and benevolent sexism, we will speak of sexism in general, to refer to ambivalent sexism. In the results shown in the following tables that relate sexism and use of specific social networks (Instagram, Snapchat, YouTube/ Blogs) in the first column of results shows the relationship of the population that scores in sexist beliefs and the percentage of people within this group that uses the specific social network, and in the second column, within the group that uses the specific social network the percentage of people who have sexist beliefs.”
And in Author’s reply they say the following:
“We clarify the difference of the first and second column in tables:2, 3 and 4, where for example: in Table 2, the % of young people who present sexism and use Instagram: refers to Instagram users within the group of sexist young people, and the % of young people who use Instagram and present sexism, refers to the group of young people who use Instagram the percentage of them who have sexist beliefs, (on page 5, lines 195-204).”
As I still do not understand it, I have operationalized what they say (I have also done a simulation in SPSS) and the group that they say they used in the study appears in the table:
|
|
sexim |
||
|
No |
Yes |
||
|
Instagramuse |
No |
|
|
|
Yes |
|
GROUP USED |
|

Author Response
REVIEWER 1:
Once again I thank you for your time and contributions to my work enriching it substantially, below I will detail the suggestions made as well as their corresponding modification (in red color and noting page and line of the text).
- Although the authors have included the number of items in the description of the questionnaires, the reliability and validity and how to calculate ambivalent sexism and the percentage of people presenting sexism remains unclear.
Two scales are used pertaining to ambivalent sexism in adolescents (DSA) and ambivalent sexism in the general population (ASI), with these being significantly correlated (p<.001). From this, scores are classified as demonstrating sexism or otherwise and the proportion of sexist responses to the two scales is calculated with a final percentage of 67.36% of adolescents reporting sexist responses (on pg.6 ; lines: 205-215).
- I still do not understand the results in Tables 3 (page 7), Table 4 (page 8) and Table 5 (Page 9). The authors have included the following text on page 5.
The second column has been removed as it generated confusion when interpreting the data. The table now presents only responses pertaining to sexism and use of a particular social network (Instagram, Snapchat, YouTube / Blogs), see tables 3,4 and 5.

Reviewer 3 Report
In this second version authors did what was suggest and improved the quality of the paper.
The objectives are in the text, but the way in which it is described is not very clear. A little modification in the writting of the objectives, the structure, can be better in order to identify it clearly.
Now, a complete reading of the whole text is need in order to check spelling or gramatical errors.
Author Response
REVIEWER 3:
Again I thank you for your time and contributions to my work enriching it substantially, below I will detail the suggestions made as well as their corresponding modification (in red color and noting page and line of the text).
- The objectives are in the text, but the way it is described is not very clear. A small modification in the wording of the objectives, the structure, may be better to identify it clearly.
We have now modified the way in which the objectives are worded. This should result in better understanding due to a better structuring of the text. (on pg. 2; lines: 73-80)
- Now, a complete reading of the whole text is needed to check for spelling or grammatical errors.
We have proofread the whole text again in order to avoid spelling and grammatical mistakes.
